# Fusion is All You Need: Face Fusion for Customized Identity-Preserving Image Synthesis

## Abstract

Text-to-image (T2I) models have significantly advanced the development of artificial intelligence, enabling the generation of high-quality images in diverse contexts based on specific text prompts. However, existing T2I-based methods often struggle to accurately reproduce the appearance of individuals from a reference image and to create novel representations of those individuals in various settings. To address this, we leverage the pre-trained UNet from Stable Diffusion to incorporate the target face image directly into the generation process. Our approach diverges from prior methods that depend on fixed encoders or static face embeddings, which often fail to bridge encoding gaps. Instead, we capitalize on UNet's sophisticated encoding capabilities to process reference images across multiple scales. By innovatively altering the cross-attention layers of the UNet, we effectively fuse individual identities into the generative process. This strategic integration of facial features across various scales not only enhances the robustness and consistency of the generated images but also facilitates efficient multi-reference and multi-identity generation. Our method sets a new benchmark in identity-preserving image generation, delivering state-of-the-art results in similarity metrics while maintaining prompt alignment.

## 1 Introduction

The synthesis of images based on text prompts has been made possible by the extraordinary capabilities of recently developed large text-to-image (T2I) models, the generated image has high quality with diverse contexts, the strong semantic prior that is acquired from a vast collection of image-caption pairs is one of the primary benefits of these models. However, the output domain of such models is restricted by the bottleneck of the quality and the number of image-caption pairs, the generated contents are hard to control only based on text prompts, and the expressiveness of text is limited, which further constrains the content generalization. Moreover, models that embed the text in an integrated language-vision space are unable to precisely reconstruct the appearance of specific subjects; they can only generate variations of the image content. Existing work like DreamBooth (Ruiz et al., 2023) aim to tackle the issue of subject inconsistent generation of T2I models, DreamBooth can be applied to different T2I models, enabling the model to produce more personalized and finely calibrated outputs after training with class-specific prior preservation loss on three to five images of a subject. However, DreamBooth need unique identifier for personalized training with specific subject and is not dedicated for human subject. IP-Adapter (Ye et al., 2023) is proposed for mitigating the restricted information expressiveness of text prompt by utilizing the image as the prompt, it is mainly based on decoupled cross-attention mechanism to separate image feature and text feature and training these newly added layers, the input to the cross attention is coming from encoders (CLIP (Radford et al., 2021) or Face embedding model) and reshaped by a projection model to match target shape if necessary. IP-Adapter can control the model generation for various content generation based on specific image prompt while ensuring the content alignment to the text, InstantID (Wang et al., 2024) is another method newly proposed for face-identity conditioned image generation also utilizing face embeddings, while previous methods such as IP-Adapter (Ye et al., 2023) and InstantID (Wang et al., 2024) achieve good image results, however, the generated images are lacking in terms of face fidelity with less controllability. Moreover, they often generate cartoonish looking faces and not completely realistic.

As illustrated in Figure 1, existing ID-preserving methods struggle to maintain consistent identity appearance across generated images. Additionally, the textual information provided in prompts is often poorly reflected in the outputs of previous methods, resulting in incomplete alignment between the prompt and the generated content. For instance, when using "Cry" style as the text prompt, models like IPA-FaceID-Plus (Ye et al., 2023) fails in terms of face fidelity and InstantID (Wang et al., 2024) fails to accurately convey the appearance of a crying face and its' results are often negatively influenced by the reference image's appearance.

In contrast, our method addresses these limitations by generating facial expressions that accurately align with the given text prompt. Specifically, we propose a novel approach that not only ensures high-fidelity face generation but also enhances identity similarity without sacrificing prompt alignment. Our method leverages the reference face image directly, overcoming the limitations of encoders like CLIP (Radford et al., 2021) by integrating the reference image with the generated one during the cross-attention process. New keys and values are introduced to capture the reference image's facial information, which is then fused with text features. This design enables our method to produce high-quality, realistic images while remaining computationally efficient for training.

Our main contributions are summarized as follows:

- We introduce a new method for enhancing ID preserving ability of existing T2I models by incorporating reference face image directly into diffusion process which achieves SOTA result in terms of ID similarity, our method shows the superiority in face controllability generation through the corresponding text prompt without degradation on face alignment.

- Our approach can also be extended for multiple identity generation and supports multiple face references.

- We conduct extensive experiments comparing our method with other SOTA identity-preserving techniques regarding facial similarity and text-image alignment, offering empirical validation for the efficacy of the proposed approach.

## 2 RELATED WORK

### 2.1 TEXT-TO-IMAGE DIFFUSION (T2I) MODELS

In recent years, diffusion models (DM) have become the state-of-the-art in text-to-image (T2I) generation, attracting significant attention on social media. Numerous studies on T2I diffusion models have been published, with continued research expected in the near future. Diffusion models can be broadly categorized into two groups based on their operational space: pixel space or latent space.

Pixel-space models, such as GLIDE (Nichol et al., 2021) and Imagen (Saharia et al., 2022), generate images directly from high-dimensional pixel data using pixel-space diffusion priors. In contrast, latent-space models, such as Stable Diffusion (Rombach et al., 2022), VQ-diffusion (Gu et al., 2022), and DALL-E 2 (Ramesh et al., 2022), operate in a compressed, low-dimensional space after the image has been encoded.

GLIDE introduced classifier-free guidance for T2I, replacing the original class labels with text prompts, which human evaluators found more effective than CLIP-guided models (Radford et al., 2021). Imagen builds on this by using pre-trained language models as text encoders and introducing dynamic thresholding for classifier-free guidance.

Stable Diffusion takes a different approach by optimizing pre-trained autoencoders to map images and their text descriptions into vectors in latent space. Cross-attention layers are employed to guide the generation in this compressed space. Similar to GLIDE (Nichol et al., 2021), it follows a classifier-free guidance approach and uses a guidance scale hyperparameter, allowing for greater alignment with the text prompt as the scale increases.

### 2.2 SUBJECT-DRIVEN IMAGE GENERATION

Subject-specific text-to-image generation utilizes a collection of photos of a specific subject to produce tailored visuals derived from textual descriptions, DreamBooth (Ruiz et al., 2023) is one of

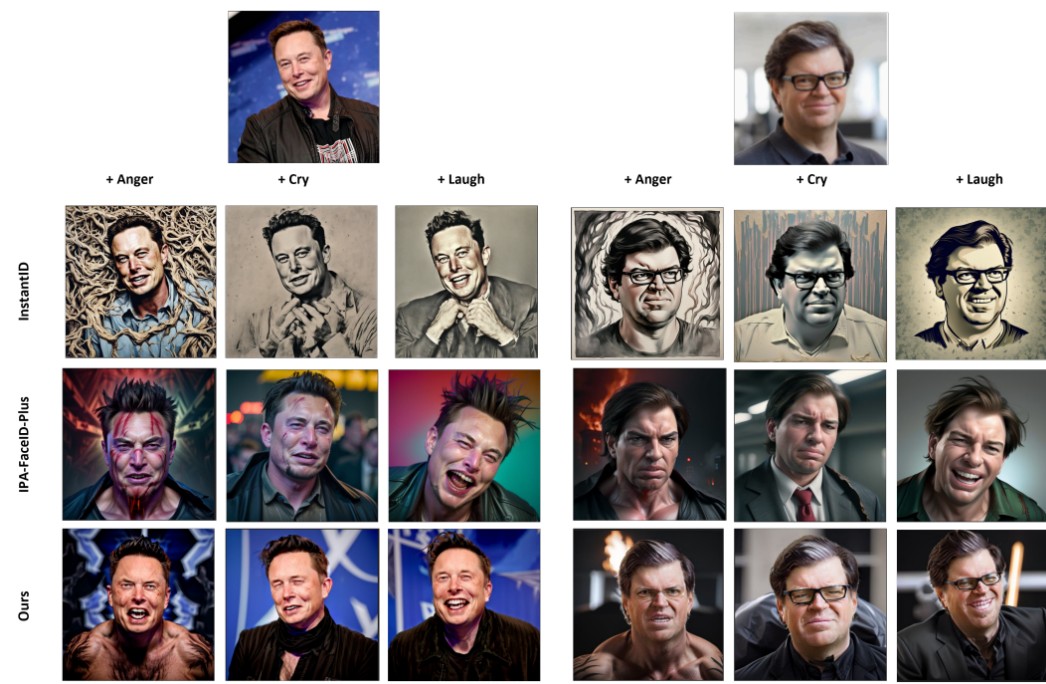

Figure 1: Comparison with other SOTA methods in terms of facial expression controllability and prompt alignment, first row are the input identities. It's worth to notice that IPA-FaceId-Plus (Ye et al., 2023) loses the face fidelity while InstantID (Wang et al., 2024) loses a bit of the prompt alignment. On the other hand our method maintains ID preservation with accurate facial expression generation.

subject-driven generation methods, that is based on fine-tuning T2I models to integrate subject information into various output context domains. DreamBooth constructs a specific prompt with a unique identifier for the pre-defined subject e.g., "A [V ] dog". Besides, the class-specific prior preservation loss is introduced for maintaining semantic information expressiveness of pre-trained model and enabling diverse subject-specific image generation. ProFusion (Zhou et al., 2023) and E4T (Gal et al., 2023) adopt a similar approach, fine-tuning a dedicated identity token for each individual. However, this necessitates separate fine-tuning for every identity, making them impractical in many scenarios.

## 2.3 ID PRESERVING IMAGE GENERATION

Current T2I models have limitations for ID-preserving image generation, the subject of the generated image is not being controlled and prompt engineering cannot promise such purpose. IP-Adapter (Ye et al., 2023) utilizes the image prompt to facilitate customized image generation. The input image is encoded into image features by pre-trained CLIP model (Radford et al., 2021), then the image features are further mapped by a linear mapping model before feeding into cross-attention layers. IP-Adapter-FaceID is based on IP-Adapter which is dedicated for realistic face image synthesis by using face ID embedding from face recognition model. For accurate portrait face generation, IP-Adapter-FaceID-Portrait is another variant which supports input multiple face images. For more realistic and ID-preserving face generation, IP-Adapter-FaceID-Plus takes directly the reference image along with face embeddings as input for enhanced face fidelity generation. Overall, all different IP-Adapter methods primarily rely on CLIP image encoder. Compared with the IP-Adapter, InstantID (Wang et al., 2024) is proposed for instant personalized image generation and is compatible with community-pre-trained models. Similar to IP-Adapter-FaceID-Plus, it relies on face embeddings with decoupled cross-attention layers, allowing the use of visual prompts. Addition-

ally, it introduces a module named IdentityNet, responsible for encoding detailed features from the reference face input image with spatial control using ControlNet (Zhang et al., 2023). Arc2Face (Papantoniou et al., 2024) is an identity-conditioned foundation model for photorealistic face image synthesis based on only the ID embedding from pre-trained ArcFace networks (Deng et al., 2019). It is proposed to generate high quality face dataset like Flickr-Faces-HQ Dataset (FFHQ) (Karras et al., 2019) but with more identities. Arc2Face uses ID embedding that extracted from a cropped face image after feeding into ArcFace model, then the ID embedding is mapped into CLIP latent space by applying text encoder and being added into cross-attention layers of fine-tuned UNet together with text features from other words, However its' primarily goal is to extract facialid features from face embeddings and not suitable for customized ID-preserving image generation.

## 3 METHOD

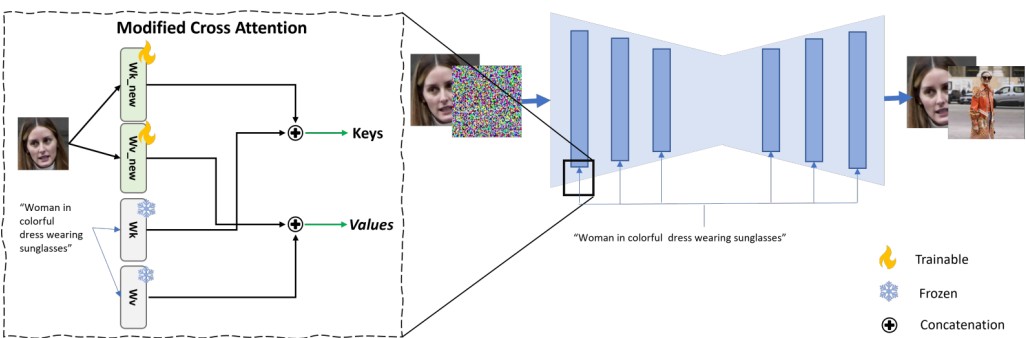

Figure 2: The overall pipeline of the proposed method. We first concatenate the target face image with the image to be generated for applying attention at different scales, we modify cross-attention layers to include new keys and values layers, their output is concatenated with the output of the original ones from text prompt.

### 3.1 OVERVIEW

We propose a novel method for customized, identity-preserving image generation with the following objectives:

- Maintain original prompt alignment: Ensure the generated images remain aligned with the original prompts across diverse image generation scenarios.

- Generalize facial features: Enable the model to generalize effectively to different face views and expressions using just a single image sample as input.

- Achieve realistic appearance and identity fidelity: Produce images that not only look realistic but also accurately preserve the facial identity of the reference subject.

- Efficient training: Design the method requires limited training resources and time.

Unlike other approaches that rely on fixed encoders (e.g., IP-Face-Adapter-plus uses a CLIP image encoder as described in (Radford et al., 2021)) or face embeddings, our method leverages the visual knowledge embedded in the pre-trained UNet of a Stable Diffusion model (Rombach et al., 2022). We input the target face identity image into the UNet, concatenated with the image to be generated. Cross-attention is then applied between these two inputs at various scales and layers within the UNet. This approach facilitates more robust and consistent generation results. To allow the use of face image in guidance, we modify the original cross-attention layers of the UNet. Specifically, we introduce new keys and values layers that take the face image as input and generate new keys and values encoding the face identity, these are then concatenated with the keys and values generated by the text layers, as illustrated in Figure 2. By avoiding the use of fixed encoders and face embeddings, which can suffer from encoding gaps, our method provides a more flexible and effective solution for identity-preserving image generation. The integration of the target face identity directly into

the UNet's cross-attention mechanism ensures that the generated images retain the desired facial characteristics while maintaining alignment with the input prompts and generalizing well across different face views and expressions.

## 3.2 FUSED ATTENTION

The cross-attention mechanism is applied at various layers of the UNet to integrate the input condition prompt c with the hidden state h, which represents the latent vector of the image being generated at a specific layer within the UNet. The queries $Q_h$, keys $K_c$, and values $V_c$ are computed as follows:

$$Q_h = Linear_q(h), K_c = Linear_k(c), V_c = Linear_v(c) \tag{1}$$

Here, $K_c$ and $V_c$ are of dimension $R^{N_c \times D}$, and $Q_h$ is of dimension $R^{N_h \times D}$.

The functions $Linear_q$, $Linear_k$, and $Linear_v$ represent the original linear layers from the Stable Diffusion (SD) model (Rombach et al., 2022). The keys and values derived from the text prompt are used to compute the attention score, which is subsequently multiplied with the values. This mechanism ensures that the guidance provided by the text is effectively integrated into the generation process. To facilitate the identity preservation and generalization capabilities, we modify this attention mechanism to incorporate the target face identity. Specifically, we introduce additional layers that process the face identity image and generate new keys and values that encode the facial features. These new keys and values are then concatenated with the original keys and values derived from the text prompt. We append two additional layers, $Linear_{k_f}$ and $Linear_{v_f}$, to the attention mechanism. These layers represent the only trainable components of our proposed pipeline. For a face image latent hidden state $h_f$ at a specific attention layer within the UNet, new keys and values that encapsulate the face identity are computed as follows:

$$K_f = Linear_{k_f}(h_f), V_f = Linear_{v_f}(h_f) \tag{2}$$

Here, $K_f$ and $V_f$ are of dimension $R^{N_h \times D}$.

The final keys $K'$ and values $V'$, which incorporate both the prompt and the face identity, are obtained by concatenating the original keys and values with the newly computed face identity keys and values:

$$K' = [K_c; K_f], V' = [V_c; V_f] \tag{3}$$

In this equation, $K'$ and $V'$ are of dimension $R^{(Nc+Nh) \times D}$.

By integrating the face identity through these additional layers and concatenation, our method ensures that the generated images not only align with the input text prompts but also accurately preserve the facial characteristics of the target identity. This approach allows for more precise control over the generated images, enabling the creation of high-quality, identity-preserving results. The efficiency of our method is maintained by limiting the trainable components to just the two additional layers, $Linear_{k_f}$ and $Linear_{v_f}$. This design choice reduces the computational overhead and training time required, making our approach practical for real-world applications. The attention scores are computed as follows:

$$S = Softmax(M + \alpha \cdot Q' K'^{T}) \tag{4}$$

Here, M denotes the attention mask, $\alpha$ is a scale term, and S represents the attention scores that incorporate both the text and the face image. The attention scores matrix S has the dimension $R^{Nh \times (N_c+N_h)}$. Finally, the output hidden state h is obtained by multiplying the attention scores with the concatenated values:

$$h = SV' \tag{5}$$

The resulting hidden state h has dimensions $R^{N_h \times D}$. As mentioned in Section 2, previous methods often separate the identity (ID) guidance into a separate attention layer, where the result of the ID attention is added to the output of the original cross-attention layer guided by the text. In contrast, we find that our approach of concatenating the keys and values from the face identity directly into the original cross-attention layer provides stronger and more consistent ID guidance. This method

avoids potential conflicts that can arise from combining two separate guiding attention layers. Additionally, our approach eliminates the need to project the face image to match a specific target shape. Instead, the concatenated keys and values are seamlessly integrated during the multiplication process, and the hidden state is restored to its original size without any additional projections.

## 3.3 MULTIPLE REFERENCES

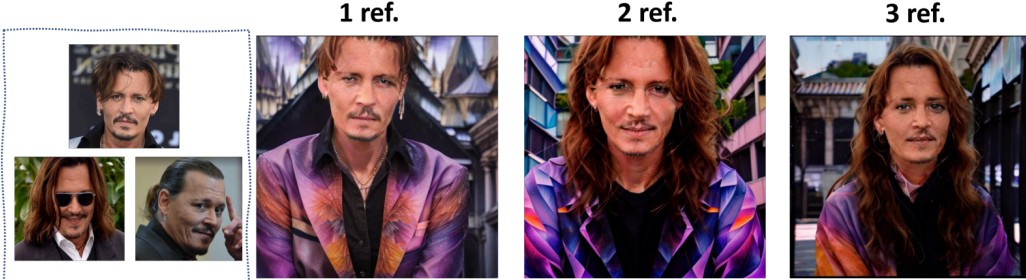

Figure 3: Comparison between using n references to guide the generation by our methods, first column represents the references identities, second column is the guidance using only one image of the references, third is using two references And fourth using all three, we notice that generated face is a mix of the features in the references.

Overall, we believe that a single image clearly showing the identity instance is sufficient for identity-preserving generation. However, there are cases where using multiple references can yield better results, especially when the face in the reference input is occluded or different features are displayed across different reference images. Our method can easily accommodate the use of multiple references. For identities $f_1, \ldots, f_n$ with corresponding hidden states $h_1, \ldots, h_n$, the keys and values can be obtained using Equation 2 and Equation 3 is then modified as follows:

$$K^{'} = [K_c; K_{f_1}; \ldots; K_{f_n}], V^{'} = [V_c; V_{f_1}; \ldots; V_{f_n}] \tag{6}$$

This modification allows for the integration of any number of identities without the need for further training or fine-tuning. We showcase sample results in Figure 3, demonstrating the effectiveness of using three identities for improved generation. The same principle can be applied to generate an image with an identity that is a mixture of distinct identities, a process commonly known as face morphing (Fu & Damer, 2022), It is important to note that the order of the identities matters, as the attention mechanism in SD (Rombach et al., 2022) does not account for this.

By extending our method to handle multiple references, we enhance the flexibility and robustness of the identity-preserving generation process. This capability allows the model to leverage information from multiple sources, leading to better handling of occlusions and variations in facial features across different images.

## 3.4 MULTIPLE IDENTITIES

Handling multiple identities in image generation remains an open challenge. In this work, we propose a solution, that combines our previously mentioned method to address this issue. The goal is to generate an image containing (n) distinct identities $f_1, \ldots, f_n$ based on input references and a given prompt. Given masks $M_1, \ldots, M_n$ where $M_i$ indicates the location mask of the face for identity i, we scale each mask and feed to each cross-attention layer in the UNet, the scaling is done based on the scale of the hidden state input to the attention and for some layer with scales s the reshaped and scaled masks are $M_1^{(s)}, \ldots, M_n^{(s)}$, for each of the face identities, we can get keys and values that are combined following Equation 6, if we applied attention like this we will obtain faces that are mix of all identities features (i.e. face morphing), we instead use a customized attention mask, denoted as $\boldsymbol{A}$, where the mask is defined as:

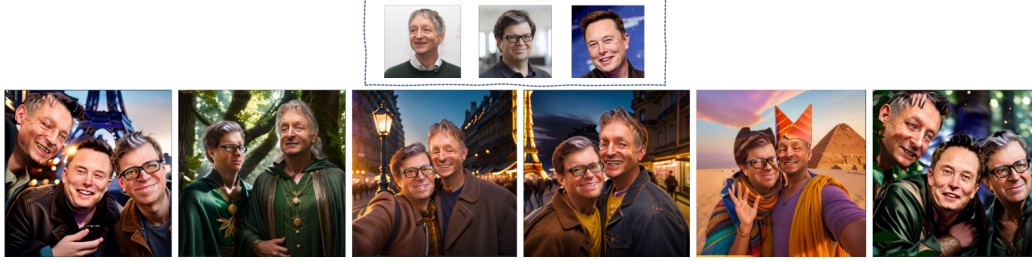

Figure 4: Multiple face identities image generation using our proposed method, first row represents the three input distinct identities, following row contains sample results for different prompts, depthmap based ControlNet (Zhang et al., 2023) was used for the pose.

$$\boldsymbol{A}_{q,k} = \begin{cases} -\inf & \text{if } isFaceKey(k) \wedge identity(k) = i \wedge \exists M_j^{(s)} : M_{j_q}^{(s)} = true \wedge j \neq i \\ 1 & \text{otherwise} \end{cases} \quad (7)$$

where q indicates the query and k is the key, this restricts the guidance for each face reference to a single, specific face, it also ensures that all identities are present in the generated image. We demonstrate the effectiveness of our method with graphical results using distinct input identities in Figure 4. These results highlight the success of our approach in generating images conditioned on multiple face identities.

## 4 EXPERIMENTS

### 4.1 EXPERIMENT SETTINGS

In our work, we utilize Stable Diffusion (Rombach et al., 2022) as the base model, specifically the Realistic Vision v4.0 checkpoint[1]. We train on a subset of the LAION-face dataset (Zheng et al., 2022), consisting of approximately 80,000 text-image pairs. All training and experiments are conducted on two Tesla V100 GPUs (32 GB), with a batch size of 1 per GPU. For comparison with other models, we use InstantID (Wang et al., 2024), which employs the SDXL model (providing an advantage over our method). For IPA-FaceId, we use IPA-FaceID-plusv2 (Ye et al., 2023), both utilizing the Realistic Vision v4.0 checkpoint as the base model. To evaluate the CLIP score, we generate a list of 100 identities conditioned on different prompts using LLMs. We provide two versions of our method, with the second version trained for more timesteps and on a larger dataset.

### 4.2 EVALUATION METRICS

Previous research on identity-preserving imagine generation tasks (Wang et al., 2024; Ye et al., 2023) lacks a defined metric for assessing identity preservation. We employ cosine distance to assess the ID-conditioning capability of various approaches in our study. Face embeddings are acquired by feeding generated images into pre-trained face recognition models. We compare the face embedding of the reference image with those from the created image; a reduced distance signifies superior ID-consistent generating performance. Additionally, we calculate the CLIP score using the CLIP model (Radford et al., 2021) to assess content alignment in face image generation across several scenarios with varied text prompts, the higher CLIP score indicates high text-image alignment. For the evaluation of image face quality, we employ SSIM and PSNR to compare the generated images with the reference images. In order to measure the face fidelity, we focus on calculating those metrics based on the face region only by cropping the faces from the generated images.

---

[1]https://huggingface.co/SG161222/Realistic_Vision_V4.0_noVAE

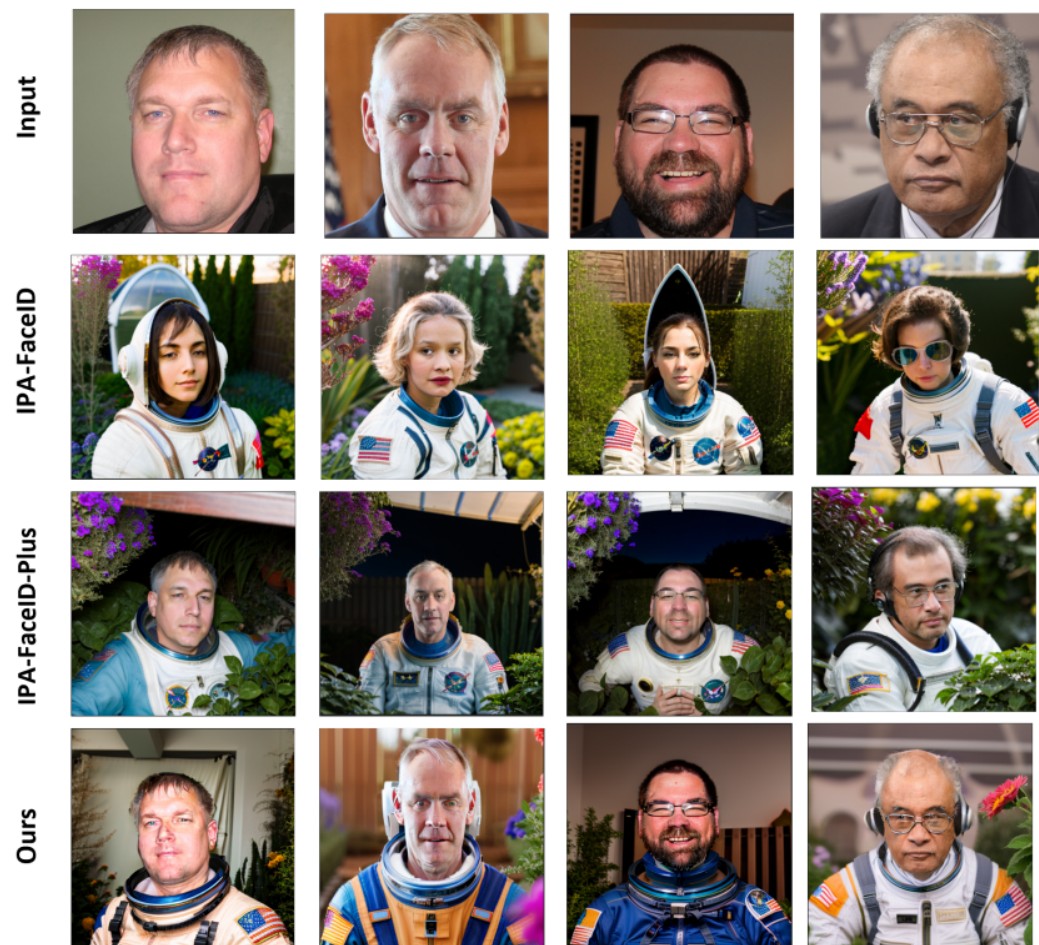

Figure 5: Comparison with other SOTA methods, with prompt used as "astronaut in a garden" and the first row represents the input target faces identities, the following rows are results from IP-faceAdapter (Ye et al., 2023), IP-face-Adapter-plus (Ye et al., 2023) and ours respectively. Identities are selected from ffhq dataset (Karras et al., 2019).

## 4.3 QUALITATIVE RESULTS

In this section, we evaluate our proposed method for visual comparison in terms of identity (ID) preservation. As shown in Figure 5, we generate images based on multiple identity face images with the same text prompt "astronaut in a garden". IPA-FaceID tends to generate face images with low ID preservation and sometimes fails to maintain the gender and age of the reference face image. On the other hand, IPA-FaceID-Plus performs better in maintaining the gender and age of the input identity; however, it still loses some original facial attributes, such as skin tone and hairstyles.

In contrast, our method generates face images with a high level of ID preservation, preserving facial expressions, gender, age, and detailed features such as hairstyles and skin color. Specifically, to investigate accurate facial expression generation, we generate images using different text prompt styles like "Anger", "Cry", and "Laugh" to control the facial emotion in the generated images. As depicted in Figure 6, InstantID generates images with less emotional alignment to the associated text prompts, while IPA-FaceID-Plus can generate the intended emotion. However, IPA-FaceID-Plus has less control over ID preservation, producing exaggerated facial expressions in random styles. Our method provides strong control of facial expressions based on the corresponding text prompt, especially in the case of the "Cry" prompt, even if the reference image already has a 'smile' or 'laugh' facial expression, our method still emphasizes the importance of the text prompt in the final

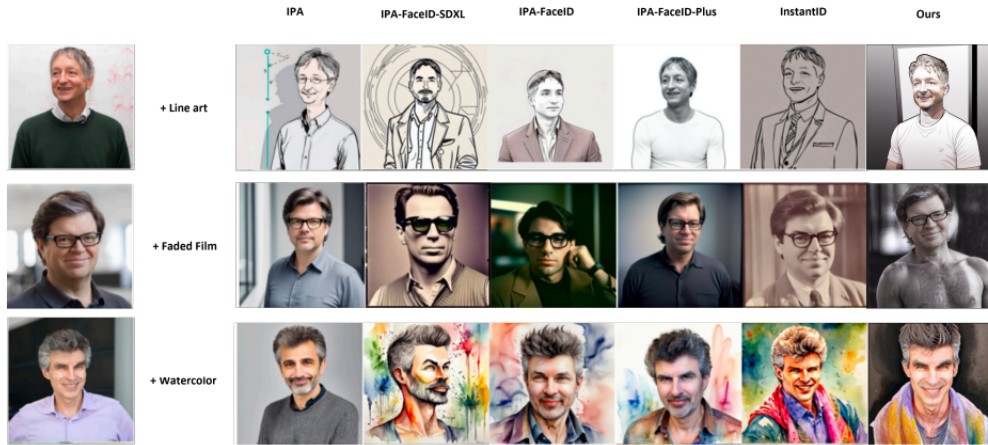

Figure 6: Visual comparison with other SOTA methods in different generation styles. The first column are the input identities along with input styles, from left to right, we show the results of IPA (Ye et al., 2023), IPA-FaceID, IPA-FaceID-SDXL, IPA-FaceID-Plus, InstantID (Wang et al., 2024) and ours. It's noticed that previous methods suffer from the lack of face fidelity or style degradation. Images are selected from InstantID (Wang et al., 2024).

Table 1: Quantative evaluations of ID preservation.

| Method | Cosine distance ↓ | | |
| --- | --- | --- | --- |
| | VGG-faces | Facenet512 | ArcFace |
| IPA-FaceID | 0.60 | 0.27 | 0.55 |
| IPA-FaceID-Plus | 0.36 | 0.25 | 0.39 |
| InstantID | 0.42 | 0.27 | 0.38 |
| $Ours_{v1}$ | 0.33 | 0.22 | 0.34 |
| $Ours_{v2}$ | **0.25** | **0.18** | **0.28** |

image generation. Our method produces ID-consistent face images while delivering more natural-looking facial expression representations, without suppressing the contribution of the text prompt to the generated images.

In Figure 6, we aim to compare the ID preservation in different image styles generation. Same as previous work InstantID (Wang et al., 2024), we use a set of text prompts to indicate various prompt styles like "Line art", "Faded Film" and "Watercolor". Our method is able to generate comparable results with other SOTA methods while maintaining high degree of face and style alignment together.

## 4.4 QUANTITATIVE RESULTS

Besides visual comparisons, we also numerically evaluate our method based on CLIP score, SSIM, PSNR and cosine distance using deepface (Serengil & Ozpinar, 2024; 2020) utilizing three models namely VGG-Face, Facenet512 and ArcFace. We utilize a subset of FFHQ (Karras et al., 2019) dataset high quality face images as the input identities in these experiments.

Table 1 shows the numerical results of cosine distance on different methods. Firstly, in terms of the face alignment, it's clear that our methods achieve significantly higher alignment than the other methods on all face recognition models. The faces in our generated image maintain high level of face features from original reference identity and can be easily used for deceiving different face recognition systems.

Table 2: Quantative evaluations of face quality and prompt alignment.

| Method | CLIP Score ↑ | PSNR ↑ | SSIM ↑ |
|---|---|---|---|
| IPA-FaceID | **25.23** | 27.94 | 0.36 |
| IPA-FaceID-Plus | 23.76 | 28.04 | 0.44 |
| InstantID | - | 27.93 | 0.33 |
| $Ours_{v1}$ | 23.28 | 28.25 | 0.41 |
| $Ours_{v2}$ | 22.51 | **28.40** | **0.47** |

In Table 2, we judge the face quality and prompt alignment offering different methods, IPA-FaceId has the highest CLIP score, however its' ID similarity is very low compared to ours and other methods, ours v1 and IPA-FaceID-Plus maintains almost similar CLIP score with IPA-FaceID-Plus being slightly better, ours v2 is worse in terms of CLIP, but it achieves significantly higher identity similarity. We exclude InstantID (Wang et al., 2024) from CLIP score due to the fact that it requires pose conditioning and the base model for the public checkpoint is SDXL based, which will make the comparison not plausible. We also compare the face quality using PSNR and SSIM metrics, our method achieves best result on both, further demonstrating that our method is capable of generating high quality realistic faces with high ID fidelity.

## 5 LIMITATIONS AND FUTURE WORK

We have observed that fine facial features tend to degrade when the face occupies a small portion of the overall image. This issue primarily stems from the limitations of Stable Diffusion (Rombach et al., 2022). Additionally, when our method is used in conjunction with ControlNet (Zhang et al., 2023), it results in slight reduction in identity fidelity, which can be resolved by light fine-tuning or reducing the conditioning strength. In our future work, we aim to address these challenges by enhancing the ability to maintain fine details and improving control in scenarios involving the generation of multiple identities.

## 6 CONCLUSION

In this paper, we present a novel method for ID-preserving image generation that excels in maintaining high identity fidelity while producing realistic facial images. Our approach demonstrates robust generalization across various facial views and expressions. Additionally, we propose innovative solutions for handling multiple references and multi-identity generation, effectively bridging encoder domain gaps and leveraging multi-reference guidance for enhanced performance. Our method achieves new state-of-the-art results in preserving identity similarity, as evidenced by our extensive experimental evaluations. Furthermore, it maintains good alignment with given prompts, setting a new benchmark in the field of identity-preserving image generation.

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
