# OpenReview forum: "FUSION IS ALL YOU NEED : FACE FUSION FOR CUSTOMIZED IDENTITY-PRESERVING IMAGE SYNTHESIS"
_ICLR.cc/2025/Conference — Submitted to ICLR 2025_

### Official Review · Reviewer_y97V · 2024-10-28

**Soundness:** 3
**Presentation:** 3
**Contribution:** 2
**Rating:** 3
**Confidence:** 5

**Summary:**

This paper is dedicated to personalized text-to-image generation. Unlike previous works, this approach not uses frozen face encoder and static facial embedding to guide the generation. Instead, they utilize the hidden states within the pretrained UNet from reference facial image to replace static facial embedding, furthermore, they choose to concatenate K, V from text and facial image.

**Strengths:**

- Personalized generation is an important task in the field of images and has not been well solved.
- It reveals a good balance between face fidelity and prompt following ability.
- It is interesting to use UNet to handle reference image instead of an extra image encoder.
- This method is simple but achieves a new SOTA in identity-preserving image generation.

**Weaknesses:**

- The novelty is kind of limited.
- Although it may be a problem of base model (SD1.5), it is hard to claim that the facial fidelity is satisfied. This is my major concern.
- The experimental results are not convincing enough, the diversity may be a big problem.

**Questions:**

- What is the attention mask in formula (4)? It is not explained here how this mask is constructed and what its role is.
- In Figure (3), the improvement brought by more reference images is not obvious.
- In this experiment, the model was trained using only 80K images, and its diversity is questionable. It should be fair to use more non-celebrity (different gender, age, ethnicity) in testing.
- In Figure (5), the generated image has a copy-paste problem. Why does this happen?
- The credibility of the quantitative results is questionable. It is hard to believe that InstantID's similarity is worse, although it does suffer from text controllability.
- Why intermediate hidden state is a better representation of face information? More discussion will be useful.

---

> ### Author Response · Authors · 2024-11-19
>
> Thanks for your comments and we reply to them below.
>
> ### Response to W1
>
> The differences between our method and IP Adapter are summarized as follows:
>
> -Flexible Guidance: Unlike IP Adapter, we do not rely on a fixed encoded representation of the reference identity image. Instead, we directly use the latent representation of the image as guidance. In contrast to IP Adapter, our guidance is not static; it varies across different layers of the U-Net, utilizing the U-Net's knowledge to apply guidance at multiple scales.
>
> -Enhanced Guidance: While IP Adapter adds a new cross-attention layer to decouple text and face identity, which may create conflicting guidance, we take a different approach. We directly modify the existing cross-attention layer by concatenating additional keys and values for attention. This process is relatively simple and dynamic, allowing for multi-reference and multi-identity guidance with ease and without the need for multiple diffusion paths; everything is achieved within a single sampling trajectory.
>
> -Image-Only Guidance: Unlike IP Adapter, which employs both face embedding and image encoding for guidance, our method relies solely on the image itself, which we found to be sufficient.
>
> Our advantages over IP Adapter and others:
>
> -Improved Identity Similarity: significant improvements across all identity similarity metrics, showcasing the superiority of our approach.
>
> -Efficiency: Our method required only 80,000 text-image pairs for training, compared to IP Adapter’s millions of pairs.
>
> -Multi-Reference and Multi-Identity Guidance: Our approach enables multi-reference and multi-identity guidance within a single sampling trajectory.
>
> ### Response to W2
> We conducted extensive experiments to evaluate ID fidelity, with the results presented in Table 1 and Table 2. These results demonstrate a significant improvement in ID fidelity across all previous methods.
>
>
> ### Response to W3
> For diversity evaluation, all our quantitative experiments use a subset of the FFHQ dataset. This subset consists of non-celebrity images and represents a diverse range of ages, races, and other attributes. For most qualitative experiments, however, we follow the approach of prior and similar works by utilizing celebrity identities, For obvious reasons.
>
> Could you elaborate on the types of experiments that might be missing when evaluating ID fidelity?
>
> ### Response to Q1
> This is the standard optional attention mask utilized by the cross-attention layers in Stable Diffusion (SD). However, it is not employed in our experiments and, by default, is also not enabled in SD.
>
> ### Response to Q2
> We believe that a single reference image is sufficient for high-fidelity generation. As demonstrated in Fig. 3, the image generated using just one reference already exhibits excellent quality. However, our goal extends beyond simply achieving higher fidelity. Instead, we aim to leverage multiple features of the same identity from different images, such as hair length, skin tone, subtle facial scars, and other fine details, to enhance the richness and versatility of the generated output.
>
> ### Response to Q3
> Same as W3
>
> ### Response to Q4
> In Figure 5, the facial expressions remain unchanged because the text prompts did not explicitly target expression modifications. This behavior is also observed in other methods, such as IPA-FaceID-Plus, though to a lesser extent, due to differences in encoding. However, as shown in Figures 1, 3, 4, and 6, our method excels in expression controllability and does not encounter such issues. We do not view this as a limitation unless it directly impacts controllability.
>
> ### Response to Q5
> Limited text controllability should not be misconstrued as an indication of better ID fidelity. InstantID primarily focuses on pose-guided generation, and the absence of ID similarity metrics in their technical report may suggest comparable results regarding ID fidelity.

---

> > ### Author Response · Authors · 2024-11-19
> >
> > ### Response to Q6
> > 1. Face Embeddings:
> > Face recognition models are not trained with a reconstruction objective, leading to a loss in embedding fidelity since the visual appearance is not fully encoded. This limitation is evident in the notable improvement of IPA-FaceID-Plus, which combines both image encoding and face embeddings, compared to IPA-FaceID, which relies solely on face embeddings.
> >
> > 2. Image Encoding:
> > Similarly, image encodings are not optimized for reconstruction or lossless compression objectives, resulting in various limitations and encoding gaps. For instance, encoders like CLIP face known constraints, as detailed in the paper "Understanding the Limitations of Text-Driven Image Models".
> >
> > Advantages of Our Approach Using Latent Vector Representation
> > 1. Lossless Reconstruction:
> > By leveraging the latent vector representation from VQ-VAE, our method benefits from the model's primary training objective of lossless reconstruction.
> >
> > 2. Enhanced Multiscale Representation:
> > The use of latent vectors allows seamless integration with the pretrained base model of Stable Diffusion (UNet). This integration enables the model to form diverse representations and understandings at different scales and layers. Applying cross-attention across these multiscale representations further enriches the fidelity and precision of the generated outputs.

---

> > ### Comment · Reviewer_y97V · 2024-11-30
> > **Response to Authors**
> >
> > Thanks for clarification.
> >
> > - Although a lower cosine distance is reported in Quantitative evaluations, the visual similarity performance is significantly lower than InstantID, making the performance of this method questionable.
> > - The training set only has 80,000 text-image pairs, and the experiments are only conducted on the FFHQ dataset. It would be more convincing if the evaluation is performed on natural faces (such as LAION-Face).

---

### Official Review · Reviewer_kuQs · 2024-10-30

**Soundness:** 3
**Presentation:** 3
**Contribution:** 3
**Rating:** 6
**Confidence:** 2

**Summary:**

This paper presents a novel text-to-image (T2I) synthesis method leveraging a modified UNet from Stable Diffusion to improve identity preservation in generated images. By altering UNet's cross-attention layers, it achieves robust multi-identity and multi-reference generation, setting new benchmarks in identity similarity metrics.

**Strengths:**

The paper presents a novel text-to-image synthesis method that excels in several key areas. By directly incorporating the target face image into the diffusion process, it significantly enhances the identity preservation capabilities of T2I models. This method consistently generates high-quality, realistic images that accurately maintain the fidelity of facial features and expressions aligned with text prompts. It also offers remarkable flexibility and scalability, supporting complex image generation scenarios involving multiple identities and reference images. Additionally, the method is designed for computational efficiency, requiring less training time and fewer resources, which makes it highly practical for real-world applications.

**Weaknesses:**

1. The method can struggle with fine facial features, especially when the face occupies a small portion of the image due to the limitations of the underlying Stable Diffusion model.

2. Potential Overfitting: The method's reliance on direct face image integration might lead to overfitting specific facial features or identities, especially in a dataset with limited diversity.

3.Could we possibly tackle the issue of facial feature degradation by improving the core architecture of Stable Diffusion or ControlNet itself, instead of just relying on the fine-tuning or adjusting the condition strength mentioned in the article?

**Questions:**

See Weaknesses

---

> ### Author Response · Authors · 2024-11-26
>
> Thank you for your comments. We address them below:
>
> ### Response to  W1
>
> We acknowledge that this issue is present in the current S1.5 version. However, there are more advanced T2I models capable of superior generation. Our method is compatible with most of these models, which minimizes this concern and makes it less critical.
>
> ### Response to W2
>
> The facial expressions remain unchanged because the text prompts did not explicitly target expression modifications. This behavior is also observed in other methods, such as IPA-FaceID-Plus, albeit to a lesser extent due to differences in encoding. Nevertheless, as demonstrated in Figures 1, 3, 4, and 6, our method significantly outperforms others in expression controllability and does not suffer from such issues. We do not consider this a limitation unless it directly affects controllability.
>
> For diversity evaluation, all our quantitative experiments utilize a subset of the FFHQ dataset. This subset includes non-celebrity images, representing a broad spectrum of ages, races, and other attributes. However, for most qualitative experiments, we follow the conventions of prior and similar works by using celebrity identities for obvious reasons.

---

### Official Review · Reviewer_4Rma · 2024-11-01

**Soundness:** 2
**Presentation:** 2
**Contribution:** 2
**Rating:** 3
**Confidence:** 4

**Summary:**

This paper proposes an identity-preserving image synthesis approach that directly passes the reference identity image to the diffusion model and fuses the attention maps obtained by a new cross-attention module with the original text-guided ones. The authors provide qualitative and quantitative results to demonstrate the effectiveness of the method.

**Strengths:**

The authors propose to pass the reference image to the diffusion model directly to avoid using an extra image encoder and potential encoding gap issues. The method can generalize to multiple reference images of the same identity and multiple identities in the same output image.

**Weaknesses:**

- **Incomplete experiments**:
  - The paper ensembles a very similar approach to IP-Adapter (referred to as IPA-FaceID-Plus in paper) with two major differences: (1) no extra image encoder, and (2) cross-attention fusion at the attention mask stage. Both are claimed by the authors to improve the performance over previous approaches. However, the authors fail to provide an ablation study for the audience to know the exact effect of these two modifications and how much they contribute to the final performance.
  - Lack of quantitative comparison with important baseline InstantID. The authors claim they use SDXL which is a superior base model over the one they use, therefore eliminated for fair comparison. But for an important baseline, the authors should either upgrade the proposed method to SDXL or downgrade InstantID to the same base model to make the comparison.

- **Unsatisfying Quality**:
Qualitative results do not show clear advances compared to previous methods. Fig.3 in particular gives very bad-quality results. Besides, Fig.5 shows signs of overfitting the reference pose and facial expression in the generated results, which raises the concern that the improved ID preservation results (in Tab.1) and improved PSNR and SSIM (in Tab.2) may be caused by this overfitting. Fig.6 does not reveal clear improvements over IPA-FaceID-Plus and InstantID also.

**Questions:**

The main concerns about the paper are listed in the Weakness section. Here are some minor questions for the authors regarding unclear writing etc.:

- Equation 4: What does Q' represent? Q value should not be modified in the proposed method based on my understanding. Also, what does M here represent? Please provide the complete formula and how it differs from the standard attention mechanism for clear understanding.
- Line 304: If you mention this technique can be used for face morphing, it is better to provide several examples to illustrate.
- Figure 4: This figure appears to be unclear. Which identities are used for each output? Why do some outputs contain three identities and some two (are those with two missing one reference or do they only get two references as input)? What depth map is used for each output?
- Section 4.1: How long does it take for training? How large is the larger dataset used to train the second version?
- Line 376: How are the face regions obtained?
- Line 427: Fig.6 has nothing to do with emotion. I guess you mean Fig.1?

---

> ### Author Response · Authors · 2024-11-26
>
> Thanks for your comments and we reply to them below.
>
>
> ---
>
> ### Addressing Weaknesses
>
> #### 1. Incomplete Experiments
>
> Regarding the differences from the IP Adapter, here is a more suitable summary:
>
> - **Flexible Guidance:**
>   Unlike the IP Adapter, we do not rely on a fixed encoded representation of the reference identity image. Instead, we directly use the latent representation of the image as guidance. Unlike the IP Adapter, our guidance is not static; it varies across different layers of the U-Net, utilizing the U-Net's knowledge to apply guidance at multiple scales.
>
> - **Enhanced Guidance:**
>   While the IP Adapter adds a new cross-attention layer to decouple text and face identity—which may create conflicting guidance—we take a different approach. We directly modify the existing cross-attention layer by concatenating additional keys and values for attention. This process is relatively simple and dynamic, allowing for multi-reference and multi-identity guidance with ease and without the need for multiple diffusion paths; everything is achieved within a single sampling trajectory.
>
> - **Image-Only Guidance:**
>   Unlike the IP Adapter, which employs both face embedding and image encoding for guidance, our method relies solely on the image itself, which we found to be sufficient.
>
> **Our advantages over the IP Adapter and others include the following:**
> - **Improved Identity Similarity:** Significant improvements across all identity similarity metrics, showcasing the superiority of our approach.
> - **Efficiency:** Our method required only 80,000 text-image pairs for training, compared to the IP Adapter’s millions of pairs.
> - **Multi-Reference and Multi-Identity Guidance:** Our approach enables multi-reference and multi-identity guidance within a single sampling trajectory.
>
> Regarding the exclusion of InstantID in quantitative experiments, the only metric not included for InstantID is the CLIP Score. This is because the reported CLIP Score for SDXL is significantly higher than that for SD1.5, making it illogical to include InstantID’s CLIP Score. Additionally, InstantID does not hold the best or even the second-best results in terms of identity fidelity, which further supports its exclusion.
>
> For CLIP score reported comparisons:
> * Dustin Podell, Zion English, Kyle Lacey, Andreas Blattmann, Tim Dockhorn, Jonas Müller, Joe Penna, Robin Rombach: “SDXL: Improving Latent Diffusion Models for High-Resolution Image Synthesis”, 2023;
>
> ---
>
> #### 2. Unsatisfying Quality
>
> - **Figure 3:**
>   We believe the identity fidelity is preserved across all references. If your concern pertains to the image quality itself, please note that quality can be affected by the prompt and seed. We encourage you to refer to other figures to evaluate overall quality.
>
> - **Figure 5:**
>   The facial expressions remain unchanged because the text prompts did not explicitly target expression modifications. This behavior is also observed in other methods, such as IPA-FaceID-Plus, though to a lesser extent due to differences in encoding. However, as shown in Figures 1, 3, 4, and 6, our method excels in expression controllability and does not encounter such issues. We do not view this as a limitation unless it directly impacts controllability.
>
> - **Figure 6:**
>   The goal of Figure 6 is not to demonstrate superior styling capability. However, we believe that our method achieves comparable styling performance while achieving higher identity fidelity, which is clearly displayed across all three identities.

---

> > ### Comment · Reviewer_4Rma · 2024-11-28
> >
> > Thank the authors for their responses. However, I don't think my concerns are fully addressed.
> >
> > \
> > For comparison with IP-Adapter, I understand the proposed method is different from and achieves superior performance over IP-Adapter. However, my concern is that the exact contributions of each different component over the IP-Adapter are missing. I believe an ablation study would be helpful.
> >
> > \
> > For comparison with InstantID, I believe it is necessary to make sure the base model is consistent across all methods. Otherwise, it is hard to say whether the performance gain is from the proposed method or the difference of base models.
> >
> > \
> > For the quality issues, noted on the explanation of undesirable quality in Figure 3. However, the identity of Johnny Depp is not fully preserved as well. For the overfitting of pose and facial expressions in Figure 5, though in Figures 3, 4, and 6 as the authors mentioned, the expression can be controlled, the identity preservation performance in those figures are worse than Figure 5. Therefore, my concern on whether the performance gain in Table 1 and 2 are caused by overfitting is not solved. In Figure 6, no clear improvement in identity preservation is observed, especially when compared with IPA-FaceID-Plus and InstantID.
> >
> > \
> > Therefore, I decide to maintain my score.

---

### Official Review · Reviewer_Smzf · 2024-11-08

**Soundness:** 2
**Presentation:** 2
**Contribution:** 2
**Rating:** 3
**Confidence:** 4

**Summary:**

This paper improves text-to-image generation by preserving individual identities in generated images. Unlike previous methods, it directly incorporates face images into the UNet of Stable Diffusion. By adjusting the cross-attention layers, the model better captures facial features across scales. This approach achieves high accuracy in both identity similarity and prompt alignment.

**Strengths:**

- This method surpasses previous approaches in identity fidelity, as evidenced by superior performance in relevant metrics.
- This method supports customizable multi-identity generation.

**Weaknesses:**

- The writing of the paper is quite rough, with at least 10 instances of incorrect punctuation usage and a generally disorganized flow of ideas.
- The paper lacks sufficient contribution, as the core method is still based on IP Adapter without original innovations. Additionally, it is challenging to observe any performance advantages.
- The results are highly limited; in Figure 4, the customized faces are barely recognizable, with significant loss of facial details. The authors need to further explain the reasons behind this. Additionally, the paper claims to address the issue of decoupling expressions from the reference image; however, in Figure 5, each result appears to be a copy-paste of the reference image’s expressions.

**Questions:**

See weekness.

---

> ### Author Response · Authors · 2024-11-13
>
> Thanks for your comments and we reply to them below.
>
> ### Response to W2
>
> The differences between our method and IP Adapter are summarized as follows:
>
> -Flexible Guidance: Unlike IP Adapter, we do not rely on a fixed encoded representation of the reference identity image. Instead, we directly use the latent representation of the image as guidance. In contrast to IP Adapter, our guidance is not static; it varies across different layers of the U-Net, utilizing the U-Net's knowledge to apply guidance at multiple scales.
>
> -Enhanced Guidance: While IP Adapter adds a new cross-attention layer to decouple text and face identity, which may create conflicting guidance, we take a different approach. We directly modify the existing cross-attention layer by concatenating additional keys and values for attention. This process is relatively simple and dynamic, allowing for multi-reference and multi-identity guidance with ease and without the need for multiple diffusion paths; everything is achieved within a single sampling trajectory.
>
> -Image-Only Guidance: Unlike IP Adapter, which employs both face embedding and image encoding for guidance, our method relies solely on the image itself, which we found to be sufficient.
>
> Our advantages over IP Adapter and others:
>
> -Improved Identity Similarity: significant improvements across all identity similarity metrics, showcasing the superiority of our approach.
>
> -Efficiency: Our method required only 80,000 text-image pairs for training, compared to IP Adapter’s millions of pairs.
>
> -Multi-Reference and Multi-Identity Guidance: Our approach enables multi-reference and multi-identity guidance within a single sampling trajectory.
>
> ### Response to W3
> Facial Detail Loss: We acknowledge the issue of facial detail loss, particularly when faces occupy a small portion of the image. This limitation is partly due to constraints within the underlying Stable Diffusion model. However, our method still achieves the best results across all identity similarity metrics, demonstrating that in terms of Facial detail, it has the best results.
>
> Identity Expressions: As shown in Figures 1 and 6, our method performs well in facial expression and style editing, successfully generating novel expressions for the input identity. In Figure 5, expressions appear unchanged due to text prompts that did not explicitly target facial expressions. Other methods, such as IPA-FaceID-Plus, also experience this limitation, albeit to a lesser extent due to encoding discrepancies. However, as demonstrated in Figures 1, 3, 4, and 6, our method does not suffer from issues with expression controllability.

---

### Meta-Review · Area_Chair_Lntf · 2024-12-17

**Metareview:**

The paper proposes an identity-preserving text-to-image generation method by modifying the UNet architecture in Stable Diffusion. Instead of using a separate image encoder, the method directly incorporates reference facial images into the cross-attention layers, aiming to improve identity fidelity and prompt alignment. However, there are several weaknesses of this paper. First, the method lacks novelty as it closely resembles IP-Adapter, experimental results are limited and unconvincing (e.g., poor facial detail and diversity). Second, key ablation studies and comparisons to strong baselines like InstantID are missing (the base model is not consistent). Additionally, qualitative results showed overfitting issues, such as copy-paste artifacts of expressions and poses. The lack of rigorous experiments across diverse datasets and incomplete analysis further weakens the submission. Last, the writing of this paper needs to be improved. While the method addresses a relevant problem and offers incremental improvements, the novelty concerns, incomplete experimental validation, and unconvincing results are significant. Therefore, I recommend rejecting the paper.

**Additional Comments On Reviewer Discussion:**

During the rebuttal, reviewers raised concerns regarding the novelty of the proposed method, its limited experimental validation, and qualitative results. Specifically, they noted the lack of ablation studies to isolate the effects of the proposed modifications (e.g., direct reference image integration and attention fusion) and the absence of fair quantitative comparisons with key baselines like InstantID. Additionally, reviewers pointed out issues with facial fidelity and diversity, with some qualitative results showing artifacts (e.g., overfitting to expressions or poses). In response, the authors clarified design choices and partially addressed the lack of ablation studies but did not provide sufficient additional experiments or fair comparisons to resolve the concerns. Qualitative improvements remained unconvincing, and claims of superior performance were not adequately supported. The final decision weighed heavily on the paper's limited novelty, incomplete experimental validation, and unconvincing results.

---

### Decision · Program_Chairs · 2025-01-22

Reject